# Aberrant CXCR4 Signaling at Crossroad of WHIM Syndrome and Waldenstrom’s Macroglobulinemia

**DOI:** 10.3390/ijms21165696

**Published:** 2020-08-08

**Authors:** Samantha Milanesi, Massimo Locati, Elena Monica Borroni

**Affiliations:** 1Humanitas Clinical and Research Center-IRCCS, Via Manzoni 56, I-20089 Rozzano (Milan), Italy; samantha.milanesi@humanitasresearch.it (S.M.); massimo.locati@humanitaresearch.it (M.L.); 2Department of Medical Biotechnologies and Translational Medicine, University of Milan, Via Fratelli Cervi, I-20090 Segrate (Milan), Italy

**Keywords:** CXCR4, WHIM syndrome, Waldenstrom’s macroglobulinaemia, signaling

## Abstract

Given its pleiotropic functions, including its prominent role in inflammation, immune responses and cancer, the C-X-C chemokine receptor type 4 (CXCR4) has gained significant attention in recent years and has become a relevant target in drug development. Although the signaling properties of CXCR4 have been extensively studied, several aspects deserve deeper investigations. Mutations in the C-term tail of the CXCR4 gene cause WHIM syndrome, a rare congenital immunodeficiency associated by chronic leukopenia. Similar mutations have also been recently identified in 30% of patients affected by Waldenstrom’s macroglobulinaemia, a B-cell neoplasia with bone marrow accumulation of malignant cells. An ample body of work has been generated to define the impact of WHIM mutations on CXCR4 signaling properties and evaluate their role on pathogenesis, diagnosis, and response to therapy, although the identity of disease-causing signaling pathways and their relevance for disease development in different genetic variants are still open questions. This review discusses the current knowledge on biochemical properties of CXCR4 mutations to identify their prototypic signaling profile potentially useful to highlighting novel opportunities for therapeutic intervention.

## 1. Introduction

The CXCL12 receptor CXCR4 is a G protein-coupled receptor (GPCR) identified for the first time in peripheral blood leukocytes [1], and initially described because of its role as a co-receptor for HIV [2]. CXCR4 is endowed with potent chemotactic properties for leukocytes but is also highly expressed in a variety of cell types, including endothelial and epithelial cells, hematopoietic stem cells, stromal fibroblasts, and cancer cells [3]. In addition to its well-established functions in hematopoiesis and immune responses, the CXCL12/CXCR4 axis plays a pivotal role in a plethora of physiological processes, including neurogenesis, cardiogenesis and neovascular formation. Receptor dysfunctions have been associated with several pathological processes, including immunodeficiencies, autoimmune diseases and cancer [4]. In particular, aberrant CXCR4 signaling caused by heterogeneous mutations affecting the C-terminal (C-ter) region of the receptor has been reported in a rare primary immunodeficiency, the Warts, Hypogammaglobulinaemia, Infections and Myelokathexis (WHIM) syndrome [5], and more recently in Waldenstrom’s macroglobulinaemia (WM), an indolent form of B-cell non-Hodgkin lymphoma [6]. These mutations act as gain-of-function mutations that turn CXCR4 into a truncated receptor with impaired internalization/desensitization and amplified signaling properties that dramatically impact on its biological functions, resulting in a “high-performance“ receptor associated with aberrant expression and activity, and with chemotherapy resistance in WM patients [7]. Since CXCR4 is an attractive target for diverse diseases and is considered as one of the best potential targets in hematological tumors, several efforts are currently ongoing aimed at developing specific inhibitors including small molecules, peptides, antibodies, and siRNA that abrogate receptor activity [8]. Of note, several clinical trials proposed CXCR4 inhibitors for treatment of hematological tumors alone or in combination with chemotherapy or biological agents [9]. Although the CXCR4 antagonist Plerixafor (also known as AMD3100) is an Food and Drug Administration (FDA)-approved drug and represents a valid therapeutic approach in both WHIM and WM, the complexity of its use in clinical practice indicates the need to develop new strategies. Other CXCR4 antagonists such as Mavorixafor (also known as AMD11070) the humanized IgG blocking monoclonal antibody BMS-936564 (also known as Ulocuplumab) are currently tested in clinical trials for WHIM and WM and showed promising results [9]. Targeting the other part of the axis, CXCL12, also represents an attractive alternative approach that may facilitate regulation rather than the abrogation of receptor activity, although the extremely high evolutionary conservation of this chemokine gets the generation of efficacious antibodies complicated. However, a selective inhibitor of CXCL12, the oligonucleotide NOX-A12, has been recently generated by Spiegelmer technology [10], and tested with promising results in clinical trials for chronic lymphocytic leukemia (#NCT01486797) [11,12] and multiple myeloma (#NCT01521533) [13], opening a new scenario in treatment of WHIM and WM. Nevertheless, only an accurate knowledge of the molecular mechanisms defining CXCR4 signaling properties will allow for developing innovative, and more targeted, therapeutic interventions. Thus, CXCR4 genetic is a crucial hub for the pathogenesis of both WHIM syndrome and WM, and deciphering the aberrant signaling profile represents a fundamental element for the identification of relevant biomarkers and the development of precision therapy in both diseases. In this review, we attempt a systematic description of WHIM mutations aimed at outlying the global organization of this group of mutations, drawing up a prototypic WHIM-mutated CXCR4 profile and identifying aspects deserving further investigation.

### 1.1. WHIM Syndrome

The CXCL12/CXCR4 axis has profound influences on immune system homeostasis, exerting a fundamental role in bone marrow (BM) colonization during ontogenesis, as well as in hematopoietic stem cells homeostasis [14]. Besides regulating BM homeostasis, CXCR4 plays a prominent function in orchestrating both innate and adaptive immune responses: it regulates leukocyte trafficking and distribution in peripheral tissues, participates in lymph node organization, and contributes to the formation and stabilization of the immunological synapse thus sustaining B cell functions as well as T cell priming [7]. Consistent with this, the prominent role of CXCR4 in immunodeficiencies has been largely documented. WHIM syndrome is a rare immunodeficiency disease caused by the combination of four main clinical manifestations: Warts, Hypogammaglobulinaemia, Infections and Myelokathexis. The first case report described a 9-year-old girl who had severe neutropenia with retention of fully mature and apoptotic neutrophils in the BM, a picture indicated as myelokathexis [15]. Several years later, the reason behind neutrophil retention/apoptosis in the BM has been linked to heterozygous mutations in the CXCR4 gene [16]. To date, 9 heterozygous mutations have been described causing WHIM syndrome [5]. All but one mutation span in the C-ter of CXCR4, thereby generating an amiss truncated receptor with delayed internalization and prolonged signaling. CXCR4 is physiologically involved in retention of neutrophils and other leukocyte subtypes in the BM, and WHIM mutations exaggerate this process causing a delay in neutrophil egress from BM to blood and enhancing neutrophil homing from blood to BM, resulting in myelokathectic neutropenia. This, in turn, increases susceptibility to bacterial and viral infections. Consistent with the gain-of-function profile of WHIM mutations, inhibitors of CXCR4 offer an opportunity to provide more specific and targeted therapy. However, this type of treatment will need to be lifelong. Plerixafor is a specific CXCR4 antagonist [17] approved by the FDA in 2008 in combination with Granulocyte Colony-Stimulating Factor (G-CSF) for hematopoietic stem cell mobilization and transplantation in patients receiving cytoreductive treatment for multiple myeloma or non-Hodgkins lymphoma (nHL) [18,19]. Efforts to repurpose Plerixafor in WHIM syndrome are underway, and to date several small studies have demonstrated its ability to reverse panleukopenia in WHIM patients.

### 1.2. Waldenstrom’s Macroglobulinaemia

The CXCL12/CXCR4 axis represents a promising prognostic marker and a potential target for therapy in oncology, as it has been shown to be operative in about 20 different cancer types where it promotes cancer cell survival and proliferation, tumor angiogenesis, and cancer cells migration to metastatic sites. It also affects chemosensitivity and disease progression by directing CXCR4-expressing tumor cells through concentration gradients of CXCL12 to reside in protective niches [19]. Consistent with this, in recent years genomic findings have provided important insights into the relevance of the CXCL12/CXCR4 axis for pathogenesis, prognostication, and treatment outcome of cancer. As CXCL12 regulates hematopoietic cell trafficking to positioning in the BM [20], it is not surprising that aberrant CXCR4 expression on cancer cells from several hematopoietic malignancies influences neoplasia progression by controlling cancer cell migration to BM as well as to lymph nodes, suggesting CXCR4 as a novel, reliable prognostic biomarker [21]. In particular, somatic CXCR4 mutations have been reported in indolent forms of B-cell nHL, follicular lymphoma, and WM [22,23]. WM is a rare form of nHL or lymphoplasmacytic lymphoma [24], characterized by lymphoplasmacytic infiltrate in BM, lymph nodes and spleen, frequently associated with the presence of monoclonal immunoglobulin M (IgM) protein in the blood. Approximately 30–40% WM patients carry mutations in heterozygosis of CXCR4 [25]. These somatic mutations are primarily subclonal, and almost always associated with the MYD88^L265P^ mutation, the first identified recurring mutation in almost 67–90% of non-IgM secreting WM patients [6]. All 17 CXCR4 heterozygous mutations identified so far in WM span in the C-ter of the receptor, and closely resemble the already documented germline mutations of CXCR4 C-ter occurring in heterozygosis in WHIM syndrome [9]. Although their relevance for clinical presentation and overall survival, as well as their relationship with resistance to chemotherapy are still unsolved issues [26,27,28], several studies reported that patients with CXCR4 mutations present a significantly lower rate of adenopathy, and those with CXCR4 nonsense mutations have an increased BM disease burden, serum IgM levels, and/or risk of symptomatic hyperviscosity [29]. Moreover, Plerixafor inhibition of CXCL12/CXCR4 axis can reverse the tumor-promoting signals of stromal cells, increasing the spontaneous apoptosis rate of tumor cells and enhancing their response to chemotherapy [30,31]. Taken together, these observations clearly demonstrate that blocking the CXCL12/CXCR4 axis might be a promising approach for potentiating the effects of currently used therapeutic regimen in WM.

## 2. Genetic Barcode of WHIM Mutations

At least 20 different WHIM/WHIM-like mutations have been described so far in the genomic region of chromosome 2q21 that encodes the C-ter of CXCR4 [9,29] (Figure 1). Interestingly, these mutations occur in the context of a gene that is highly conserved across species and span in a genomic region that is even more highly conserved than the gene as a whole [32].

In WHIM syndrome, all but one of the CXCR4 germline mutations truncate receptor C-ter by premature termination (4 NS) or by frame-shifts (4 FS) that introduce from 3 to 24 additional new amino acids [5]. The only missense mutation (MS) is E343K, which involves a single amino acid substitution and a charge change [32]. By far the most common WHIM mutation is R334X, which accounts for half of genotyped cases in the literature, including both de novo and familial cases. Conversely, CXCR4 S338X accounts for 15 of the genotyped cases [33]. The remaining 29 genotyped cases are distributed among the remaining mutations, 8 of which were observed in only 1–3 cases each.

Acquired CXCR4 WHIM-like mutations, are present in up to 40% of patients with WM and are nearly always observed in conjunction with the MYD88^L265P^ mutation [29]. CXCR4 mutations are essentially unique to WM, as they have not been described so far in other malignancies, except for several cases of patients affected by follicular lymphoma [9,22]. Among the different CXCR4 mutations observed, 5 are NS and 12 FS [34]. The most frequent mutations are the S338X (C1013G) followed by S338X (C1013A), which generate a stop codon in place of a serine at amino acid position 338. Several other mutations to S338 has been described, suggesting a hotspot locus in the CXCR4 gene. The S338 hotspot mutation has been identified in half of CXCR4 mutated cases in WM. Interestingly, WM acquired exact the same R334X mutation that has been also previously described in WHIM syndrome. Differently from MYD88^L265P^, CXCR4 mutant clonality is highly variable, and multiple CXCR4 mutations are present within individual patients in separate clones or are present as heterozygous events [29]. The subclonal nature suggests that these mutations are acquired after MYD88^L265P^, although this could occur early in WM pathogenesis.

## 3. Signaling of WHIM Mutations

Like all GPCRs, the structure of CXCR4 contains 7 transmembrane helices, an extracellular N-terminal domain and an intracellular C-ter domain. CXCR4 activity, trafficking, and signaling properties are finely coordinated by physical interaction with several canonical (i.e., G proteins, GPCR kinases (GRKs), and β-arrestins) and non-canonical proteins (i.e., Filamin A (FLNA), Atrophin–Interacting Protein 4 (AIP4), CD74 and CD164) [35]. CXCR4 is known to couple to the G protein αi (Gαi) that mediates most of receptor signaling pathways, and to Gα12–13 and Gαq [36]. These signaling pathways include the activation of Src and PI3K kinases, and Phospholipase C (PLC) that in turn activates Protein Kinase B (PKB also known as AKT) and Protein Kinase C (PKC)/ Mitogen-Activated Protein Kinase (MAPK), respectively. AKT and PKC are also activated by the interaction of CXCR4 with the Human Leukocyte Antigen (HLA) class II histocompatibility antigen gamma chain (CD74) [37] and Endolyn (CD164) [38]. After CXCL12 activation, Gαi is negatively regulated by powerful desensitization mechanisms that involve GRKs, which phosphorylates specific serines and threonines distributed along the C-ter to which β-arrestins are recruited to drive receptor internalization [39,40]. CXCR4/β-arrestins interaction is finely tuned by FLNA which stabilizes the receptor at the plasma membrane by blocking receptor endocytosis [41], unlike AIP4 which promotes receptors ubiquitination and targeting to multivescicular bodies for degradation [42]. By linking CXCR4 to actin, FLNA also orchestrates the deep cytoskeletal rearrangements required to sustain cell migration, like other non-canonical constitutively interacting proteins such as Drebrin [43], diaphanous-related formin-2 (mDIA2) [44], PI3-kinase isoform p110γ [45] and the motor protein NonMuscle Myosin H Chain (NMMHC) that selectively bind the receptor C-ter domain [46].

To date, all but one WHIM CXCR4 variant show impaired internalization, with prolonged receptor residence time on the cell surface which is thought to contribute to amplification and prolongation of receptor signaling activity [47]. Thus, paradoxically, a loss of structure from WHIM mutations leads to a gain-of-function by stabilizing the mutant receptor on the cell surface (Figure 2).

The signaling profile of WHIM-mutated CXCR4 has been currently explored for 5 mutations (the NS mutations R334X, G336X, S338X; the MS mutation E343K; the FS mutation L329fs341X) (Table 1), whereas signaling properties of remaining variants are presently largely unknown. To date, R334X and S338X are the best characterized mutations, and S338X is the mutation whose signaling properties has been investigated for both WHIM and WM diseases. Studies have been conducted either in ex vivo setting using freshly isolated leukocytes (PBMC) or CD34+ cells from patients, and immortalized cells (EBV [48,49], BCWM.1 [50], MWCL-1 [51] for WHIM and WM, respectively), and in in vitro experimental settings using several cell lines (Table 1).

### 3.1. The Gain-of-Function Nature of WHIM Mutations

The binding of CXCL12 to CXCR4 takes place through a two-step mechanism [64]. CXCL12 contact at the extracellular domain causes a first conformational change which strengthens chemokine binding to a receptor pocket. Next, a second conformational change activates the intracellular trimeric G protein by the dissociation of Gαi subunit from the Gβ/Gγ dimer at the third receptor intracellular loop [65,66]. Once activated, the Gαi subunit inhibits production of cAMP by adenyl cyclases and stimulates the activity of Src family tyrosine kinases that modulate cell cycle progression by activating the Ras/Raf/MEK/ERK pathway. Conversely, Gβγ and Gα subunits stimulate the activity of PI3Ks, which mediate gene transcription, cell adhesion and migration, and cell survival by phosphorylating AKT and several focal adhesion components. AKT also triggers the activity of PLC that hydrolyses PIP2 into IP3 and DiAcylGlycerol (DAG), which promote Ca^2+^ mobilization from the intracellular stores and PKC/MAPK activation, respectively.

With the rest of GPCR structure left intact, WHIM-mutated CXCR4 remains fully competent in downstream signaling via G protein. Impaired internalization prolongs receptor residence time on the cell surface and contributes to amplification and prolongation of receptor signaling that have been observed using multiple signaling assays in different cellular contexts [47]. To date, increased ERK and AKT activation have been described for R334X and S338X, and amplified calcium flux and chemotaxis has been proven for all mutations but not yet reported for S338X and L329fs341X, respectively (Table 1). Interestingly, polymerization of actin monomers into F-actin filaments has been also evaluated in T lymphocytes of R334X and S338X WHIM patients, with results showing a protracted rise of F-actin after CXCL12 stimulation [33]. These observations are consistent with the “hyperactive” signaling nature of WHIM-mutated CXCR4 and might reflect an increased ability of WHIM variants to activate G proteins. However, effect of WHIM mutations on G protein recruitment and activation deserves further investigation. Lagane and colleagues measured CXCL12-induced 35S-GTPγS binding to activated Gαi-containing membranes from HEK293 cells expressing similar amounts of CXCR4 WT and S338X, and reported that the truncation of the C-ter improved CXCL12-induced coupling efficiency and potency of the receptor [61], confirming previous observations by Balabanian and colleagues on S338X and R334X using the same in vitro assay [33]. Thus, the enhanced responsiveness of WHIM-mutated CXCR4 to CXCL12 is likely the consequence of an improved activation of receptor-associated G proteins.

### 3.2. Hyperactivity of WHIM-Mutated CXCR4: A Question of TAIL or deTAILs?

Upon activation, GPCRs are rapidly phosphorylated, typically by members of the GRK family. This triggers the recruitment of β-arrestins at receptor C-ter, which prevents further activation of their cognate G protein, initiates β-arrestin-dependent signaling, and leads to receptor desensitization and internalization [39,40]. The CXCR4 C-ter contains 18 potential phosphorylation sites: 15 serines and 3 threonines. The “phosphorylation barcode” of the CXCR4 C-ter includes 7 serine residues that are phosphorylated by specific GRKs and PKC isoforms following CXCL12 stimulation: S321, S324, S325, S330, S339, a residue between S346 and S348, and either S351 or S352 (Figure 1) [67]. CXCL12 induces long-lasting GRK2/3-dependent phosphorylation of a S346/347 phosphosite which precedes less stable phosphorylation at S324/325 and S338/339 phosphosites [68]. S330 and S339 are phosphorylated by GRK6, as well as S324/325 that are additionally phosphorylated by PKC [67]. The S346/347 residues have been shown to be rapidly phosphorylated by GRK2, GRK3 and PKC(ɑ) following CXCL12 stimulation and together with the SHSK motif in intracellular loop 3 (ICL3) are involved in the recruitment of β-arrestin2 [69], which initiates the process of receptor desensitization [68,70]. Of note, β-arrestin2 binding to these two sites has distinct functional consequences; whereas β-arrestin2 interaction with the C-ter mediates receptor desensitization and endocytosis, its binding to ICL3 is important for β-arrestin2-mediated signaling [69].

The hierarchy of CXCR4 phosphorylation events may partially explain how small structural changes caused by WHIM mutations are leveraged into large functional effects in CXCR4. All WHIM mutants lacking the S346/347 phosphosite have impaired phosphorylation at the intact proximal S324/325 and S338/339 sites, and this is consistent with impaired CXCL12-induced receptor downregulation from the cell surface (Table 1). However, although the MS E343Kmutation shows impaired phosphorylation at serine phosphosites and results in increased receptor signaling, the effect on blocking normal receptor downregulation is impaired [32]. Similarly, the NS S336X resulted in enhanced migration, but calcium flux and receptor downregulation were unaffected [56]. Phosphorylation of CXCR4 C-ter is mandatory for docking of β-arrestins, which also serves as scaffold for a number of downstream effectors involved in G protein-independent signal pathways, enhancing receptor-mediated ERK and p38 activation which promote cell migration [71]. However, the role of β-arrestins and GRKs in orchestrating the puzzling behavior of WHIM mutations is still an unsolved and controversial issue. McCormick and colleagues demonstrated in HeLa cells that R334X is defective at recruiting β-arrestin2 and GRK6 proteins after exposure to CXCL12, and displays a delay in receptor downregulation, signaling and trafficking in comparison to WT [55]. These authors proposed that truncated residues of CXCR4 C-ter are important for the interaction with GRK6 but not GRK3, providing evidence that different GRKs may require distinct CXCR4 structural elements for interaction. This was consistent with GRK6 depletion results that showed a delayed CXCL12-induced ERK activation, similarly to that seen downstream R334X. Conversely, a study by Lagane and colleagues in leukocytes isolated from WHIM patients carrying the S338X mutation demonstrated that β-arrestin2 interaction with CXCR4 is preserved, despite that the receptor is not internalized, and results in an enhanced ERK activation and consequent chemotaxis to CXCL12 [61]. Furthermore, the authors demonstrated that β-arrestin2 also bind the SHSK motif of ICL3, with prolonged signaling downstream WHIM receptor mutants. Of note, the observation that CXCR4 C-ter and ICL3 (LKTTVILIL) interact with FLNA (Figure 1), an important regulatory element of the cytoskeleton known to act as a platform for receptor signaling and intracellular trafficking [72], that not only links CXCR4 to the actin cytoskeleton, but also suggests its role in shaping receptor signaling by regulating β-arrestin2/ICL3 interaction [41]. Gomez-Mouton and colleagues demonstrated that FLNA also binds the ICL3 of R334X and suggested a role both in receptor stabilization at the cell surface and in increased β-arrestin2 binding to the mutant, providing a new molecular mechanism for hyperactivation of WHIM receptor signaling [41]. Interestingly, a detailed study of two unrelated patients with a full clinical form of WHIM syndrome but lacking detectable mutations in CXCR4 showed hyperactive responses to CXCL12 [33]. Balabanian and colleagues identified a defective activity of GRK3 as a consequence of decreased GRK3 levels, likely resulted from defective mRNA synthesis [62]. As GRK3 specifically regulates CXCL12-mediated desensitization and internalization of CXCR4, its defective activation possibly contributes to the amplification of the G protein-dependent responses in WHIM cells as shown by gene transfer mediated expression of GRK3 in this patient’s fibroblasts or leukocytes that recovered normal chemotactic response and CXCR4 internalization in response to CXCL12 [62]. These results underline the complexity of CXCR4 modulation by WHIM mutations, and clearly demonstrate that the critical biochemical feature shared by WHIM patients carrying C-ter truncating mutations of CXCR4 and those who lack detectable mutations in CXCR4 is the gain in functional hyperactivity of CXCR4 in response to CXCL12. This further supports the general understanding of WHIM syndrome as an immunodeficiency caused by functional hyperactivity of CXCR4. Thus, while investigating the molecular basis accounting for altered CXCR4 signaling in WHIM patients, possible functional alterations of components normally involved in the CXCR4 signaling machinery have emerged.

### 3.3. Effect of Homo or Heterodimerization on WHIM-Mutated CXCR4 Signaling: A Still Unanswered Question

In the past two decades, an increasing amount of evidence suggested that beyond monomeric entities, GPCRs could also exist and function as oligomeric complexes [73], thus further adding complexity to the signaling basis of chemokine-mediated responses. CXCR4 was shown to constitutively dimerize, with CXCL12 binding further enhancing receptor homodimerization [4]. Furthermore, CXCR4 form heterodimers with other chemokine receptors, including its atypical chemokine receptor counterpart ACKR3 (also known as CXCR7). Although the biological significance of CXCR4/ACKR3 heterodimerization remains poorly understood, it has been shown that by this mechanism ACKR3 acts as key regulator of CXCR4 signaling properties, including its ability to mobilize intracellular Ca^2+^ [74,75,76,77]. Once homodimerized, CXCR4 becomes phosphorylated through the rapid binding and activation of Janus Kinase 2 (JAK2) and 3 (JAK3) mediated by a G protein-independent pathway [78]. JAK proteins in turn activate the STAT signaling involved in regulation of various cellular processes, including Ca^2+^ mobilization from intracellular stores and the transcription of target genes [4]. However, despite the emerging interest in this pathway, the role of JAK/STAT in CXCR4 signal transduction is still matter of debate.

As WHIM mutations occur in heterozygous dominant condition, the WT and WHIM-mutated CXCR4 alleles are likely co-expressed. The crystal structure of CXCR4 revealed that the receptor is a homodimer [79,80], thus both the WT and WHIM-mutated forms could coexist as independent monomers and/or as homodimers and/or as heterodimers in patient cells. Although at present the biochemical evidence that WHIM-mutated variants can heterodimerize with the WT receptor is limited [61], this element has potential unpredictable effects on receptor expression levels, G protein/β-arrestin coupling and signaling properties. The stoichiometry of the different forms could also vary according to cell type and among different patients, possibly contributing to the phenotypic heterogeneity observed in different patients with the same pedigree. Moreover, it has recently been reported that in addition to the ligand-mediated conformational change of the receptor that activates G proteins, CXCR4 undergoes changes associated with receptor nanoclustering that are necessary for signaling and could establish a new target for potential intervention in WHIM and WM patients [81]. A deeper understanding of the functional and pharmacological properties of CXCR4 heterodimers may therefore indicate more specific targets for drug design and development.

## 4. Concluding Remarks

After the initial description of the presence of mutations in the CXCR4 gene in both WHIM and WM diseases, several studies have revealed a striking concordance in the molecular and biochemical properties of the underlying mutations, suggesting that hyperactivation of receptor signaling by truncation of the C-ter is an essential aspect in pathogenesis. Biochemical studies have provided support for the model that impaired CXCR4 downregulation and desensitization leads to the characteristic immunologic and hematologic alterations, although the broad heterogeneity in clinical manifestations and disease severity and different abnormalities in immunological parameters in the two diseases tempt us to speculate that they are likely supported by different underlying mechanisms. Since genetic studies have also identified patients with the same clinical features but without mutation of CXCR4, uncharacterized downstream regulators of the receptor may also be involved in a proportion of cases. Though the CXCR4 inhibitor AMD3100 is an FDA-approved drug and represents a valid therapeutic approach in WHIM and WM, issues related to the complexity of its use in clinical practice indicate the need to develop new therapies. These will require a more precise understanding of signaling properties of CXCR4 mutants and the role they may play in WHIM and WM pathogenesis.

## Figures and Tables

**Figure 1 ijms-21-05696-f001:**
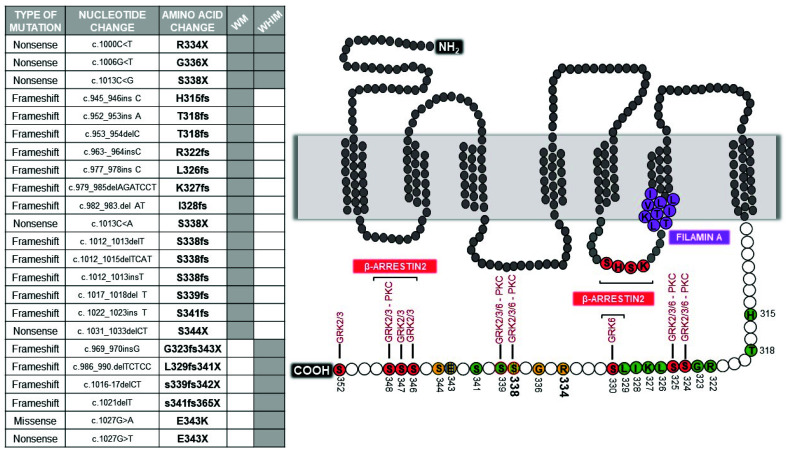
Mutations in C-ter of CXCR4. Mutations identified in WHIM and WM patients are listed in the table. The schematic structure of CXCR4 reports sites of mutation, with frameshift mutations highlighted in green, nonsense mutations in yellow, and missense mutations in blue. Phosphorylation sites for GRKs and PKC, and docking sites for β-arrestin2 are filled in red. The most frequent S338 and R334 mutation sites are represented in bold.

**Figure 2 ijms-21-05696-f002:**
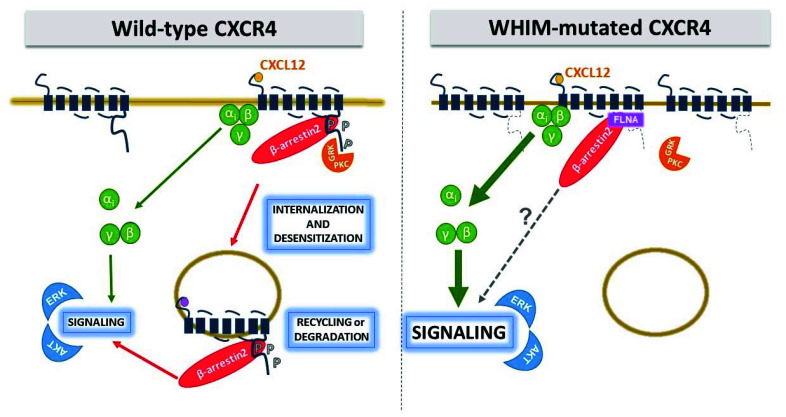
Gain-of-function nature of WHIM-mutated CXCR4. WHIM mutations cause a gain of G protein-dependent functions, at least in part by abrogating normal β-arrestin-mediated receptor downregulation. The C-ter is the site of receptor phosphorylation sites by GRKs and PKC and their loss affects β-arrestin binding and receptor internalization. P, phosphate.

**Table 1 ijms-21-05696-t001:** Biological, biochemical and signaling properties of WHIM-mutated CXCR4 variants.

Type of Mutation	G PROTEIN	β-ARRESTIN	Calcium	ERK	AKT	CXCR4 Internalization	CXCR4 Membrane Expression	CXCR4 Desensitization	CXCR4 Eterodimerization	Cell Migration	Ibrutinib Response	Disease and Cell Type	Reference
**R334X**												**WHIM:** K562, CHO	[52]
												**WHIM:** CD34^+^, K562	[53]
												**WHIM:** EBV	[16]
												**WHIM:** NK	[54]
												**WHIM:** HeLa, HEK293	[55]
**G336X**												**WHIM:** PMBC	[56]
**S338X**											**R**	**WM:** BCWM.1, MWCL-1	[57]
											**R**	**WM:** BCWM.1	[58]
											**R**	**WM:** Patients (undefined)	[59]
											**R**	**WM:** Patients (undefined)	[60]
												**WHIM:** PBMC, HEK293	[61]
												**WHIM:** PBMC, CHO	[33]
												**WHIM:** EBV	[62]
**E343K**												**WHIM:** K562, PBMC	[32]
**L329fs**												**WHIM:** K562, PBMC, HEK293	[63]

Evidence for unmodified, increased or impaired activity of WHIM-mutated CXCR4 compared to WT are shown in blue, green, and red, respectively. Weak impairment is indicated as light red. White boxes indicates absence of data. Properties reported have been evaluated upon CXCL12 stimulation in the indicated cellular contexts of WHIM and WM diseases. R, resistant.

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
