# Peer review of "Aberrant CXCR4 Signaling at Crossroad of WHIM Syndrome and Waldenstrom’s Macroglobulinemia"

_ijms, 2020, doi:10.3390/ijms21165696_

Round 1
Reviewer 1 Report
It is a very well-organized review, especially for gain-of-function mutations.
However, the authors' group recently published the following review article.
Vaccines (Basel)
2020 Apr 3;8(2):E164. doi: 10.3390/vaccines8020164.
New Insights on the Emerging Genomic Landscape of CXCR4 in Cancer: A Lesson From WHIM
Stefania Scala, Crescenzo D'Alterio, Samantha Milanesi, Alessandra Castagna, Roberta Carriero, Floriana Maria Farina, Massimo Locati, Elena Monica Borroni.
Here too, the CXCL12/CXCR4 axis plays a crucial role in influencing cancer biology, promoting survival, proliferation, angiogenesis, and inducing migration of cancer cells to metastatic sites. It is a prognostic marker and therapeutic target. Most recently, a CXCR4 C-terminal mutation has been identified in the genomic environment of patients affected by the rare B-cell tumor, Waldenstrom macroglobulinemia. These mutations are associated with warts, hypogammaglobulinemia, immunodeficiency, and myelocatoxys (WHIM) syndrome, aberrant expression and activity of CXCR4, and immunodeficiency associated with chemoresistance in clinical trials. Very similar. This review provides a summary of current knowledge of the relevance of CXCR4 mutations in cancer biology, focusing on their clinical manifestations and their importance as predictors of response to treatment.
So please clarify a bit more about what the two reviews look like (citing them) and how they differ.
Author Response
We thank the reviewer for his/her comment. CXCL12/CXCR4 axis has gained significant attention in the recent years because of its pleiotropic functions, including its pivotal role in hematopoiesis and immune responses, as well as in cancer by controlling cell survival and proliferation, tumor angiogenesis, metastasis, chemosensitivity and disease progression. Germline mutations in CXCR4 have been firstly identified in a rare congenital immunodeficiency known as WHIM syndrome. The C-terminal region of CXCR4 harbored these mutations that are associated with receptor aberrant expression and activity and with chemotherapy resistance in clinical trials. More recently, somatic WHIM-like mutations have been firstly identified in the genomic landscape of cancer patients affected by Waldenstrom's Macroglobulinaemia, a rare B cell non-Hodking lymphoma. Unfortunately, a paucity of knowledge for these mutations in cancer exists because they have not been previously described in any malignant condition despite the often-reported dysregulation of CXCR4 gene expression in several cancers. Thus, taking advantages from WHIM’s teachings we undertook a challenge to collect, summarize and integrate the current knowledge on the relevance of CXCR4 mutations in cancer biology in two different reviews that approach this topic from different points of view.
Our first review (Scala S, Vaccines. 2020) focuses on the role of CXCL12/CXCR4 axis in cancer and the genomic landscape of CXCR4 providing the first summary list of the mutations that have been so far identified in cancer patients through a detailed analysis of literature and database repositories. Given the biological relevance of CXCR4 and the crucial impact of its genomic landscape in the pathogenesis of Waldenstrom's Macroglobulinaemia and Diffuse Large B-Cell Lymphoma, two rare B cell non-Hodking lymphoma, the review finally focuses on the clinical and therapeutic importance of CXCL12/CXCR4 axis in these hematological malignancies and explores the impact of WHIM-like mutations as predictors of clinical presentation and response to therapy. Moreover, the ongoing pharmacological strategies targeting of CXCR4 in hematological malignancies are discussed and the review delves into the topic in more detail.
Conversely, our second review in IJMS journal focuses on the molecular mechanisms responsible of the aberrant signaling of WHIM-like mutations, a topic that in the Vaccines review have been only mentioned in a short paragraph in Appendix A “CXCR4 genetic on WHIM”. Herein, IJMS review provides a more concise description of the role of CXCL12/CXCR4 axis and WHIM-mutations in both WHIM and Waldenstrom's Macroglobulinaemia compare to Vaccines review, but attempts a detailed, systematic description of the biochemical/signaling properties of WHIM mutations and their biological relevance aimed at outlying the global organization of this group of mutations. It was challenging to collect and summarize all data in literature and further discuss, but in our opinion the effort we have done represents the strength, originality and uniqueness of this review compare to the others on the same topic. We exploited the impact of the mutations on both G protein and β-arrestin signaling modules, as well as on homo/heterodimerization properties of the receptor, drawing up a first prototypic WHIM-mutated CXCR4 profile potentially useful to highlighting novel opportunities for therapeutic intervention and identifying aspects deserving further investigation.
Reviewer 2 Report
It’s a very interesting and well written review. I have only the following minor revision.
- Since accumulating evidence indicates that CXCR4 receptors are regulated by both interacting proteins and post-translational modifications, it could be useful to introduce an additional subchapter related to (physical) interaction with canonical and non canonical interacting proteins. Please introduce in the manuscript the subchapter and relative references.
- Although the CXCR4 inhibitor AMD3100 is an FDA-approved drug and represents a valid therapeutic approach in WHIM and WM, issues related to the complexity of its use in clinical practice indicate the need to develop new therapies. These will require a more precise understanding of signaling properties of CXCR4 mutants and the role they may play in WHIM and WM pathogenesis. In this review it could be briefly discussed. A number of tools have been developed to block CXCR4/CXCL12 interactions, and they are currently under different stages of development. Most of the tools reported target the CXCR4 receptor, since chemokines are generally viewed as ‘undruggable’ proteins. Recently ghave been proposed tools that bind to the ligand, CXCL12, rather than the receptor. Notably, targeting a specific ligand may facilitate regulation rather than the elimination of receptor activity.
Author Response
We thank the reviewer for his/her comments.
Here a
1- Since accumulating evidence indicates that CXCR4 receptors are regulated by both interacting proteins and post-translational modifications, it could be useful to introduce an additional subchapter related to (physical) interaction with canonical and non canonical interacting proteins. Please introduce in the manuscript the subchapter and relative references.
We agree with the reviewer that an additional subchapter related to the physical interaction of CXCR4 with canonical and non-canonical interacting proteins could be useful to improve the manuscript. We introduced this topic and relative references in the manuscript by integrating the first paragraph of Section 3 (“Signaling of WHIM mutations”) that already discusses the signaling properties of CXCR4 with the information we collected from literature. A more detailed description of canonical and non-canonical interacting proteins that has been described involved downstream WHIM mutations has been already reported in the subsequent subparagraphs (i.e. G proteins in “The gain-of-function nature of WHIM mutations”; GRKs, β-arrestins and FLNA in “Hyperactivity of WHIM-mutated CXCR4: a question of TAIL or deTAILs?”).
2- Although the CXCR4 inhibitor AMD3100 is an FDA-approved drug and represents a valid therapeutic approach in WHIM and WM, issues related to the complexity of its use in clinical practice indicate the need to develop new therapies. These will require a more precise understanding of signaling properties of CXCR4 mutants and the role they may play in WHIM and WM pathogenesis. In this review it could be briefly discussed. A number of tools have been developed to block CXCR4/CXCL12 interactions, and they are currently under different stages of development. Most of the tools reported target the CXCR4 receptor, since chemokines are generally viewed as ‘undruggable’ proteins. Recently ghave been proposed tools that bind to the ligand, CXCL12, rather than the receptor. Notably, targeting a specific ligand may facilitate regulation rather than the elimination of receptor activity
We agree with the reviewer that the complexity of the use of Plerixafor in clinical practice suggests the need to develop new therapies to targeting CXCR4. Thus, a more detailed understanding of signaling properties of CXCR4 mutants and their role in WHIM and WM pathogenesis represent a crucial hub for the development of precision therapy in both diseases. As suggested by reviewer, we briefly discussed this aspect in Section 1 (“Introduction”).
Reviewer 3 Report
The review paper of CXCR4 signaling concerning to WHIM syndrome and WM is very interesting and meaningful. Overall, it is well written. The signal per se is complicated, so I guess it was challenging to summarize the previous results and further discuss.
I have a few minor comments as follows.
Page 4, Line 147, Gαi and Gα appear in the text and figure. What is the difference?
Page 4, Line 157, remaining variants are largely unknown. What are the remaining variants? The authors listed 5 mutations in Table 5, so I wonder what “remaining” means?
Page 5, Line 190, rinse of F-action, what does rinse mean?
Page 7, Line 264 caosed should be caused?
Author Response
We thank the reviewer for his/her comments.
1- Page 4, Line 147, Gαi and Gα appear in the text and figure. What is the difference?
We missed Gαi in the figure. We provide a new corrected version of Figure 2.
2- Page 4, Line 157, remaining variants are largely unknown. What are the remaining variants? The authors listed 5 mutations in Table 5, so I wonder what “remaining” means?
Remaining variants refer to the other identified mutations in C-ter of CXCR4 beyond the 5 indicated in Table 1 that are listed in the table embedded in Figure 1. We provide a modified version of the sentence.
3- Page 5, Line 190, rinse of F-action, what does rinse mean?
We made a spelling mistake: the correct word is rise. We provide a corrected version of the sentence.
4- Page 7, Line 264 caosed should be caused?
Yes, we made a spelling mistake: the correct word is caused. We provide a corrected version of the sentence.